# Novel Giant Phages vB_AerVM_332-Vera and vB_AerVM_332-Igor and Siphophage vB_AerVS_332-Yulya Infecting the Same *Aeromonas veronii* Strain

**DOI:** 10.3390/v17081027

**Published:** 2025-07-22

**Authors:** Igor V. Babkin, Vera V. Morozova, Yuliya N. Kozlova, Valeria A. Fedorets, Artem Y. Tikunov, Tatyana A. Ushakova, Alevtina V. Bardasheva, Elena V. Zhirakovskaya, Nina V. Tikunova

**Affiliations:** Laboratory of Molecular Microbiology, Institute of Chemical Biology and Fundamental Medicine Siberian Branch, Russian Academy of Sciences, Novosibirsk 630090, Russia; morozova@niboch.nsc.ru (V.V.M.); ulona79@mail.ru (Y.N.K.); f.valeriya41@gmail.com (V.A.F.); arttik@ngs.ru (A.Y.T.); ushakova@niboch.nsc.ru (T.A.U.); herba12@mail.ru (A.V.B.);

**Keywords:** Caudoviricetes, giant phages, phiKZ-like, siphoviruses, *Aeromonas veronii*, comparative genomics

## Abstract

Three novel *Aeromonas* phages vB_AerVS_332-Yuliya, vB_AerVM_332-Vera, and vB_AerVM_332-Igor and their host *Aeromonas veronii* CEMTC7594 were found in the same water + sediments sample collected in a freshwater pond. Complete genome sequencing indicated that vB_AerVS_332-Yuliya (43,584 bp) is a siphophage, whereas vB_AerVM_332-Vera (294,685 bp) and vB_AerVM_332-Igor (237,907 bp) are giant phages. The host strain can grow at temperatures from 5 °C to 37 °C with an optimum of 25–37 °C; siphophage vB_AerVS_332-Yuliya effectively reproduced at temperature ≤ 25 °C, the optimal temperature for giant phage vB_AerVM_332-Igor was 25 °C, and giant phage vB_AerVM_332-Vera infected host cells at 5–10 °C. The genomes of these phages differed significantly from known phages; their level of nucleotide identity and values of intergenomic similarity with the corresponding neighboring phages indicated that each of these phages is a member of a new genus/subfamily. Giant phage vB_AerVM_332-Vera is a member of the proposed *Chimallinviridae* family, which forms Cluster D of giant phages that possibly evolved from phages with shorter genomes. Giant phage vB_AerVM_332-Igor is part of Cluster E, the known members of which preserve the size of genomes. Phages from Cluster F, containing *Aeromonas* phages among others, show a gradual decrease and/or increase in genomes during evolution, which indicates different strategies for giant phages.

## 1. Introduction

Bacteriophages are an essential part of the biota on Earth. Along with eukaryotic viruses, phages are the most common microorganisms and play an important role in ecology and maintaining the bacterial balance in various natural niches [1]. Bacteriophages are characterized by a significant diversity in the organization of their genomes and virions. At the very beginning of virological studies, it was believed that bacteriophages have small sizes of viral particles and the largest of them is *Enterobacteria* phage T2 with its head-to-tail length varying from 211 nm to 248 nm, depending on the nutrient media [2]. However, *Bacillus megatherium* phage G with its head-to-tail length of 600 nm was discovered in 1969 [3] and in 1978, *Pseudomonas aeruginosa* phage phiKZ with a particle size exceeding 300 nm was found [4]. In the 21st century, the first double-stranded DNA genome with a length of 280 kb of the phage phiKZ was sequenced and analyzed [5]. Phages with genomes of more than 200 kb were named giant or jumbo phages [6,7,8], whereas phages with genomes of more than 500 kb, megaphages [9]. Currently, ~850 phage genomes of >200 kb are deposited in the NCBI GenBank (https://www.ncbi.nlm.nih.gov/genbank, accessed on 30 September 2024).

The currently known giant phages show significant divergence in their genomes. There is ongoing debate about the origin of giant phages and two hypotheses have been proposed: they have independently evolved from smaller phages or they have an ancient origin from a common large ancestor and developed their own replication systems within each group [10,11,12,13]. It has recently been shown that the structure of the phage phiKZ capsid is complex, with many proteins involved in its formation [14]. Probably, such complex capsids are inherent in most, if not all, phiKZ-like phages [15]. Moreover, it has been found that at least several studied phiKZ-like giant phages form nucleus-like structures in infected cells. In the ongoing evolutionary battle between bacteria and phages, certain giant phages develop a nucleus-like structure that isolates their replicating genomes preventing host defense factors from accessing them. Phages from the *Chimalliviridae* family encase their DNA in a compartment made of a protein called chimallin. Chimallin is self-assembles as a flexible sheet. This compartment blocks ribosomes and separates transcription from translation. PhiKZ-like phages carefully control which proteins enter the phage nucleus and which remain in the bacterial cytoplasm [16,17,18,19]. Considering these facts, it is difficult to explain how such complicated structures could be formed during the evolution of giant phages from small progenitors.

Giant phages have been isolated from various sources around the world and they infect different hosts among Gram-negative and Gram-positive bacteria [10,20,21]. These phages have been often found in aquatic environments [22,23] and it has been assumed that the aqueous medium facilitates the diffusion of large phage particles [24]. Of all genomes of giant phages, at least 53 genomes (>6%) belong to phages infecting *Aeromonas* spp. Notably, efforts have been mainly aimed at isolation and characterization of phages infecting *Aeromonas hydrophila* and *Aeromonas salmonicida* that are significant for healthcare and fishing industry [21,25,26,27].

Here, two novel genomes of giant phages vB_AerVM_332-Vera and vB_AerVM_332-Igor (henceforth referred as 322-Vera and 322-Igor) infecting the same *Aeromonas veronii* strain are described. One of the phage is a phiKZ-like phage, whereas the other one has no related phages assigned to a known family. In addition, the third siphophage vB_AerVS_332-Yulya infecting the same bacterial strain was isolated from the same location as these giant phages and the *A. veronii* strain. These three phages reproduce at very different rates; however, the optimal temperature for their reproduction varies. This coexistence of three phages parasitizing a single host demonstrates the possibilities of evolutionary adaptation of phages to limited environmental resources, including even slow-growing giant phages.

## 2. Materials and Methods

### 2.1. Bacterial Strains and Growth Conditions

The strain *A. veronii* CEMTC 7594 that was used as the host for phage identification and propagation was isolated from a water + sediment sample collected from a small fresh pond located in the city of Sirius, Krasnodar Territory, Russia. This sample was agitated and one gram of the agitated sample was used to isolate bacterial strains. Insoluble particles were pelleted (2000× *g*, 3 min) and tenfold dilutions of the obtained supernatant, which were prepared in sterile PBS, pH 7.5 were plated onto Nutrient agar (NA) plates (Microgen, Obolensk, Russia). Then, the plates were incubated at room temperature for 24 h. To select individual strains, the grown bacterial colonies were spread on another NA plates and cultivated at the same conditions. The procedure was repeated at least three times and morphology of bacterial cells was observed. The taxonomic position of each of the selected clones was determined by sequencing a fragment of the 16S rRNA gene (1308 bp) using primers 8F 5′-AGRGTTTGATCCTGGCTCA-3′ and 1350R 5′-GACGGGCGGTGTGTACAAG-3′ as described previously [28]. The selected strains were deposited in the Collection of Extremophile Microorganisms and Type Cultures (CEMTC) of the Institute of Chemical Biology and Fundamental Medicine SB RAS, Novosibirsk, Russia.

To induce prophages, two methods were used. First, an exponentially growing culture of the *A. veronii* CEMTC 7594 (the OD600 value of 0.4) was treated with mitomycin C, 0.5 μg/mL (Sigma-Aldrich, St. Louis, MO, USA) and then incubated with shaking for 24 h at 25 °C. The OD600 value was measured every hour to record growth or possible lysis. Second, a fresh layer of *A. veronii* CEMTC 7594 in the top agar was exposed to ultraviolet irradiation for 5, 10, and 15 s. Then, plates were incubated for 42 h and the appearance of plaques was regularly checked. The assays were carried out with three replicates.

The host strain and other *Aeromonas* strains from CEMTC were cultivated in Nutrient broth (Thermo Fisher Scientific, Waltham, MA, USA) or plated onto NA (Microgen, Obolensk, Russia).

### 2.2. Phage Isolation

The same water + sediment sample from a pond was used for phage isolation. This sample was centrifuged at 7000× *g*, 12 °C for 15 min and the supernatant was filtered using a 0.22 µM pore size membrane syringe-driven filter (Millipore, Guyancourt, France) and screened for phages using bacterial stains isolated from the same sample. For this purpose, plates with fresh layers of the isolated bacterial strains in the 0.8% top agar were prepared and 10 µL aliqots of sterilized supernatant was spotted on them and icubated at 25 °C overnight. As plaques were observed on a plate with a layer of the *A. veronii* CEMTC 7594 strain, this strain was considered as a host strain. Individual plaques were cut out, suspended in 100 µL of sterile PBS, and the suspension was incubated with shaking for 16 h at RT to extract phage particles. Then, tenfold dilution aliquots of the phage suspension were plated onto the top agar containing the *A. veronii* CEMTC 7594 cells and plates were incubated overnight at 25 °C to obtain single phage plaques for subsequent phage extraction. The cycle of phage plating and extraction was repeated three more times. To isolate DNA of giant phages, phage suspension was obtained from the first plaque without subsequent cloning.

When the genomes of two additional giant phages were discovered, the same method with modifications was used to isolate giant phages. The host strain *A. veronii* CEMTC 7594 was inoculated with a phage suspension obtained from the first confluent plaque and five aliquots of the mixture were immediately placed at different temperatures (5 °C, 10 °C, 18 °C, 25 °C, or 37 °C) for phage adsorption (without shaking) and further cultivation (with shaking) at the corresponding temperatures. Then, serial dilutions of the infected cultures, obtained from different temperatures, were spotted on fresh layers of the host strain in the 0.3%, 0.5%, and 0.8% top agar and icubated at the corresponding temperature overnight (at 18 °C, 25 °C, and 37 °C) or 48 h (at 5 °C and 10 °C).

### 2.3. Phage Propagation

The exponentially growing *A. veronii* CEMTC 7594 culture was inoculated independently with the obtained phage suspensions and cultivated at 25 °C overnight to propagate phage/phages. When cell lysis appeared, bacterial debris was pelleted (10,000× *g*, 10 min, RT) and phage particles were precipitated from supernatants by adding polyethylene glycol 6000 (PEG 6000; AppliChem, Darmstadt, Germany) with NaCl as described previously [29].

### 2.4. Biological Properties and Host Range Assay

To perform experiments on the kinetics of cell culture lysis, *A. veronii* CEMTC 7594 was cultivated until OD600 = 0.5 at 25 °C, infected with the phage at MOI of 0.05 and allowed to adsorb at different temperatures (at 5 °C, 10 °C, 18 °C, 25 °C, or 37 °C) for 30 min without shaking. Then, infected cultures were incubated at the corresponding temperature with shaking. Aliquots were taken every hour (up to 48 h), diluted, and spread on the NA plates. The plates were incubated at the corresponding temperature. Bacterial colonies were counted and statistical analysis and graphs were prepared using GraphPad Prizm v. 8.0 (https://www.graphpad.com, accessed on 15 November 2024). All experiments were performed three times in each repeat.

The host range was determined as described previously [30]. Briefly, serial dilutions of phage suspensions were added to freshly prepared bacterial lawns in top agar. A total of 77 *Aeromonas* strains belonging to ten species were applied (Appendix A). Strains deposited in CEMTC ICBFM SB RAS were used for this experiment.

### 2.5. Phage DNA Isolation

All DNA isolation procedures were performed independently for two samples of phage suspensions—obtained from the first plaque and from the phage plaque after cloning. To isolate phage DNA, pellets that were obtained after PEG-precipitation [29] were dissolved in STM buffer (10 mМ NaCl, 1 mМ MgCl_2_, and 50 mМ Tris-HCl, pH 8.0) and the suspensions were incubated with 5 µg/mL of RNase and DNase (Thermo Fisher Scientific, Waltham, MA, USA) for 1 h at 37 °C. Then, 0.5% SDS, 20 mM EDTA, and proteinase K, 200 mkg/mL (Thermo Fisher Scientific, USA), were added and the mixtures were incubated for 3 h at 55 °C. Phage DNA was extracted using a phenol–chloroform mixture and the DNA was twice precipitated by adding 2.5 volumes of 96% ethanol.

### 2.6. Phage DNA Sequencing

Phage DNA from both samples was fragmented by sonication for 40 sec on ice, using the ultrasonic Sonopuls HD 2070 homogenizer (Bandelin, Berlin, Germany) and used for preparation of pair-end libraries with the NEBNext Ultra II DNA Library Prep Kit for Illumina (New England Biolabs, Inc., Ipswich, MA, USA). Sequencing was performed using the MiSeq Benchtop Sequencer and MiSeq reagent kit v2 (2 × 250-cycles) (Illumina, San Diego, CA, USA). The obtained reads were filtered by quality using Trimmomatic v.0.32 (https://github.com/usadellab/Trimmomatic, accessed on 29 September 2024). SPAdes v.3.15.4 (https://github.com/ablab/spades, accessed on 29 September 2024) was used for phage genomes de novo assembly.

### 2.7. PCR and Real-Time PCR for Phage Genomes Detection

Phage genomes were detected in *A. veronii* cultures using PCR. Pairs of primers specific to each genome are listed in Table 1. To perform PCR, each pair of primers was independently mixed with the tested DNA and the mixture was added to the PCR-basic kit (SibEnzyme, Novosibirsk, Russia). GeneAmp PCR System (Applied Biosystems, Waltham, MA, USA) was used in accordance with the manufacturer’s protocol.

Real-time PCR was performed to quantify the ratio of phage genomes in bacterial cultures after infection with a mixture of bacteriophages. *A. veronii* CEMTC 7594 cells were infected with a suspension containing a mixture of phages and aliquots of the infected culture were cultivated at 11 °C, 18 °C, 25 °C, and 37 °C. Total DNA isolated from the samples was used as a template for real-time PCR with primers and probes indicated in Table 1. Each DNA sample obtained from infected bacteria, which were cultivated at a certain temperature, was diluted 10, 100, 1000, and 10,000 times. Each of these dilutions was used in three monoplex reactions with corresponding primers and probes. DNA isolated from A. veronii CEMTC 7594, also grown at different temperatures but not infected with phages, was used as a control. Thus, a total of 96 monoplex reactions were carried out, each in three technical replicates. PCR was carried out in a total volume of 25 μL, the reaction mixture contained 2.5 μL of 10x SE-Buffer (SibEnzyme, Novosibirsk, Russia), 2.5 μL of dNTPs, 0.2 μL of TaqSE DNA polymerase, 5 U/μL (SibEnzyme, Novosibirsk, Russia), and 11.8 μL of deionized water. Then, 1 µL of each oligonucleotide (forward primer, reverse primer, and probe, 10 pM each) and 5 µL of DNA template were added. CFX96 Touch Real-Time PCR Detection System (Bio-Rad, Hercules, CA, USA) was used with the following protocol: 95 °C/5 min followed by 50 cycles of 95 °C/10 s, 55 °C/10 s, 60 °C/20 s, with the fluorescence detection at the end of elongation.

### 2.8. Phage Genome Analysis

PhageTerm v.1.0.12 software [31] (https://galaxy.pasteur.fr/?tool_id=toolshed.pasteur.fr%2Frepos%2Ffmareuil%2Fphageterm%2FPhageTerm%2F1.0.12&version=1.0.12&__identifer=q05ix39bvq; accessed on 20 August 2024) was applied to find the position of the phages termini. VectorBuilder’s GC Content Calculator (https://en.vectorbuilder.com/tool/gc-content-calculator.html, accessed on 24 September 2024) was used to determine the GC content of the phage. To analyze genomes, the Ori-Finder software was used [32] (https://tubic.org/Ori-Finder, accessed on 22 September 2024). Phage genomes were searched for the tRNA genes using tRNAscan-SE software [33].

Putative open reading frames (ORFs) were determined and annotated using Rapid Annotation Subsystem Technology (RAST) v.2.0 [34] (https://rast.nmpdr.org, accessed on 12 August 2023). The identified ORFs were verified manually using BLASTX algorithms against nucleotide and protein sequences, deposited in the NCBI GenBank (https://ncbi.nlm.nih.gov, accessed on 23 September 2023). In addition, the InterProScan [35], HHPred, and HMMER tools [36] were applied for the identification of hypothetical proteins. To edit and align the nucleotide sequences, BioEdit [37] and MAFFT [38] (https://mafft.cbrc.jp/alignment/server, accessed on 30 September 2023) tools were used. The search for core genes was carried out using the Coregenus program (https://coregenes.ngrok.io/, accessed on 15 November 2024).

### 2.9. Phylogenetic Analysis

Comparative proteomic analysis was carried out using ViPTree [39] (https://www.genome.jp/viptree, accessed on 20 August 2024) and VIRIDIC tools [40] (http://rhea.icbm.uni-oldenburg.de/VIRIDIC, accessed on 15 September 2024). The phylogenetic analysis of the essential proteins encoded by the genomes was carried out as follows: the most similar protein sequences identified by the BLASTP search were extracted from the NCBI GenBank; then, the sequences were aligned and analyzed using MEGA 11.0 [41]. Mega 11 was used to build phylogenetic trees using Maximum Likelihood estimation with 1000 bootstrap replicates.

## 3. Results

### 3.1. Identification and Characterization of the Host

Several strains of aerobic bacteria of different genera were isolated from a sample containing water and sediment taken from a small freshwater pond near the Black Sea in the city of Sochi (Krasnodar Territory). Only one of them, *A. veronii* CEMTC 7594, was found to be a host for phages isolated from the same sample. To determine the optimal growth temperature for the *A. veronii* CEMTC 7594 strain, cells were grown at various temperatures (5 °C, 10 °C, 18 °C, 25 °C, or 37 °C). Bacterial growth was monitored by measuring the optical density at 600 nm (OD600). For *A. veronii* CEMTC 7594, the OD600 value of 1.0 corresponded to 8 × 10^8^ cells/mL. The obtained results indicated that this strain could grow at all specified temperatures, including 5 °C (Figure 1); however, the optimal growth temperature for this strain varied between 25 °C and 37 °C, as the stationary growth stage occurred after 10 h at both temperatures (Appendix A). This result is consistent with the previously obtained data that *A. veronii* cells are able to grow in a wide temperature range with an optimum between 22 °C and 35 °C [42].

To check whether the *A. veronii* CEMTC 7594 genome contains a prophage encoding a functional lysogenic phage, mitomycin С and ultraviolet irradiation were used to induce prophage. Both methods were used in three repeats; however, possible prophages were not induced.

To isolate phages/phage, all bacterial strains found in the same water + sediment sample were used as potential hosts; however, plaques were found only on a plate with a layer of the *A. veronii* CEMTC 7594 strain (Figure 1). Plates were incubated at 10 °C and 25 °C and small clear plaques were observed at 10 °C, whereas two types of plaques were revealed at 25 °C—large plaques with a halo and small clear plaques (Figure 1A). To check if different plaques are formed by different phages, both large and small plaques were cloned independently and tested in A. veronii CEMTC 7594 at 10 °C and 25 °C. However, phage particles independently isolated from small and large plaques formed, in turn, both small clear plaques and large plaques with a halo at 25 °C. The isolated phage was named vB_AerVS_332-Yuliya (332-Yuliya).

Lytic activity of the phage was assayed at 5 °C, 10 °C, 18 °C, 25 °C, and 37 °C. Multistep bacterial killing curves indicated that the number of live host cell decreased by four orders of magnitude six, three, or two hours after infection at 10 °C, 18 °C, and 25 °C, respectively; the phage was low active at 5 °C and inactive at 37 °C (Figure 1B).

For the host range assay, 77 *Aeromonas* strains from ten species were used (Appendix A). Most of the strains were isolated from rivers and natural water reservoirs; 23 strains were determined as *A. veronii.* The obtained results indicated that the phage 332-Yuliya infected only the host strain *A. veronii* CEMTC 7594 and another strain *A. veronii* CEMTC 7445 from a nearby pond was not sensitive to this phage, which indicates that this phage has a narrow host range.

### 3.2. Sequencing of Phage Genomes

Two pair-end libraries were constructed and sequenced. The first library was constructed using DNA of a phage propagated after four times cloning of the initial plaque in *A. veronii* CEMTC 7594. The vB_AerVS_332-Yuliya (332-Yuliya) phage genome was assembled de novo as the double-stranded DNA circular contig with the average coverage of 196. The genome size was 43,584 bp (Figure 2).

Taking into consideration that giant phages are often discovered in aquatic environments [22,23], the second pair-end library was constructed based on the DNA of phage particles isolated from the initial plaque and reproduced in the same host strain. Indeed, two additional circular genomes of giant phages were assembled in addition to the 332-Julia genome. These phages were named vB_AerVM_332-Vera and vB_AerVM_332-Igor (322-Vera and 322-Igor) and their genomes were double-stranded DNA, with sizes of 294,685 bp and 237,907 bp, respectively (Figure 3). The average coverage for the 332-Yuliya, 322-Vera, and 322-Igor genomes was 98, 56 and 71, respectively. The obtained genome sequences genomes were deposited in the NCBI GenBank database under accession numbers PQ474292 (332-Vera), PQ474294 (332-Igor), and PQ474293 (332-Yuliya).

### 3.3. Phage Genomes Organization

Analysis of genome sequences indicated that the 332-Yuliya, 322-Vera, and 322-Igor genomes differ significantly. Obviously, there were differences in the lengths and number of predicted ORFs (Table 2). The GC content of the 332-Yuliya, 332-Vera, and 332-Igor genomes was different (Table 2) and lower than that for the host species (~59%). The GC content of the 332-Yuliya genome (~48%) was more than ten percentage points less than the host one, which suggests significant differences in codon usage between the phage and host cells. Analysis of DNA termini and probable phage packaging mechanisms by both PhageTerm and Li’ methods showed the absence of obvious termini in both giant phages, while the 332-Yuliya phage genome contained direct terminal repeats (DTRs), yielding its physical size of 43,584 bp (Table 2). These data suggest a T7-like strategy for DNA packaging of the 332-Yuliya phage [43], whereas 332-Vera and 332-Igor probably use a headful packaging mechanism without a certain packaging signal [44]. The Ori-Finder program predicted with high reliability the position of the origin of replication (ori) in the Julia 332 genome (Appendix A); however, the position of ori in the studied genomes of giant phages was predicted with low reliability, since many important elements of ori were not detected (Appendix A). The search for the tRNA genes showed their absence in the genomes of all three phages. Genes responsible for the lysogenic lifestyle were not found in the genomes of these phages, which indicates their lytic nature (Appendix A).

### 3.4. Phage Genomes Detection by PCR and Real-Time PCR

Considering that phages 332-Yuliya, 322-Vera, and 322-Igor capable of infecting a single host, PCR was performed using appropriate primers (Table 1) and total DNA isolated from the culture of *A. veronii* CEMTC 7594 grown at different temperatures, as a template. PCR fragments, both specific and non-specific, were not detected. The obtained results indicated that the genomes of isolated phages are absent in the form of prophages in the host strain.

As these three phages were found in the sample, and none of the phages were lysogenic, attempts were made to obtain these phages individually. The host strain was inoculated with a phage suspension obtained from the first confluent plaque and five aliquots were immediately placed at different temperatures (5 °C, 10 °C, 18 °C, 25 °C, or 37 °C) for phage adsorption (without shaking) and further cultivation (with shaking) at the corresponding temperatures. Plates with fresh layers of the host strain in the 0.3%, 0.5%, and 0.8% top agar were prepared and serial dilutions of the above mixtures applied to the plates with different agar in parallel. Plates were cultivated in the corresponding temperatures. A total of 400 plaques obtained on different top agar at various temperatures were tested by PCR with specific primers (Table 1). As a result, only the 332-Yuliya phage was detected.

Due to the difficulties associated with the separation of phages in culture, it was decided to detect their genomes using real-time PCR with specific primers and probes. Cq values averaged over technical replicates with standard deviation, as well as data on amplification efficiency are shown in Table 3. A quantification threshold was set in the exponential phase of all amplification curves. Amplification efficiency was calculated based on a standard curve constructed from the results of the samples dilutions series. For all samples the amplification efficiency exceeded 90%; the average efficiency was 95.8%. All negative controls with DNA from uninfected bacteria, cultivated at the same temperatures, did not demonstrate an increase in fluorescence.

The obtained results indicated that siphophage 332-Yulia propagated at temperatures from 10 °C to 25 °C with the same efficiency; at 5 °C, efficiency of its propagation was lower and at 37 °C, only traces of the 332-Yulia DNA were observed (Table 3). Both giant phages reproduced in a narrower temperature range: phage 332-Igor had an optimum growth at 25 °C, whereas phage 332-Vera only multiplied weakly at low temperatures (5 °C and 10 °C) (Table 3). Obviously, the lack of a positive result in the separation of three phages was due to a significant difference in the number of virions of these phages in the infected culture.

### 3.5. Comparative Analysis of the 332-Yuliya Genome

Two main clusters of genes were found in the 332-Yuliya genome (Figure 2). The first cluster contains mainly structural genes located behind the genes of the terminase subunits in the forward direction. In this cluster, the genes encoding portal protein, major capsid protein, tape measure protein, tail terminator protein, and others. Notably, the tailspike protein gene is located in the second cluster containing genes oriented in the reverse direction (Figure 2). The genes encoding DNA polymerase and four transcriptional regulatory proteins were detected in this cluster. The RNA polymerase gene was not found in the 332-Yuliya genome (Appendix A).

Comparative ViPTree analysis indicated that 332-Yuliya is a member of a group of unclassified phages with siphovirus morphology and the most similar phage is the *Aeromonas* siphophage vB_Aves_KLEA5 (Figure 4). Pairwise comparisons of the 332-Yuliya genome with two most similar ones confirmed the results obtained using the ViPTree analysis (Figure 4B). However, despite the Aves_KLEA5 genome shows notable similarity with 332-Yuliya, the genes encoding endolysins and tail spike proteins of these phages have no similarity at al.

Phylogenetic analysis of products of the signature genes shown that the terminase large subunits of 332-Yuliya was in a well-supported branch with that of vB_Aves_KLEA5 phage (Figure 4C); however, major capsid proteins of these phages were distant (Figure 4D).

Based on the analysis of the 332-Yuliya genome, it was concluded that this phage belongs to the yet unclassified family of siphophages. The level of nucleotide identity (NI) of the coincidence was 69% between the 332-Yuliya and *Aeromonas* vB_AveS_KLEA5 genomes and 38% between the 332-Yuliya and *Vibrio*_phage_Artemius genomes; the values of intergenomic similarity determined in the VIRIDIC program were 69.4% and 9.0%, respectively. The obtained NI level indicates that both *Aeromonas* phages are close to being assigned to the same genus as the level of 70% was established by ICTV Bacterial Virus Subcommitee for creating phage genera [45].

### 3.6. Comparative Analysis of the 332-Vera Genome

Comparative genome analysis indicated the 332-Vera genome belongs to a large complex group of giant phages (Figure 5A). This group contains a wide variety of giant phages with genome sizes from 201 kb to 321 kb that infect various Gram-negative bacteria and are members of the new family *Chimallinviridae*.

Genes encoding DNA polymerase B and two multisubunit RNA polymerases (msRNAP), which are typical for phages from the *Chimallinviridae* family [46,47], were found in the 332-Vera genome. One of the msRNAPS is virion RNA polymerase (vRNAP), which is packaged in a capsid and injected into the cell along with DNA for transcription of the early genes; this enzyme is formed by five subunits. Non-virion RNA polymerase is produced during infection and is formed by four subunits. All subunits of two msRNAP polymerases were detected in the genome of the 332-Vera phage (Figure 3A, Appendix A).

Also, the signature genes of phages from the *Chimallinviridae* family are the genes encoding chimallin and tubulin [18,48,49] and both were identified in the 332-Vera genome (Figure 3A, Appendix A). Chimallin forms a nucleus-like structure that protects the phage genome from bacterial anti-phage defense systems and separates transcription from translation. In turn, tubulin is assembled into special filaments that can translocate the nucleus-like structure inside the infected cell and ensure the transport of procapsids to the nucleus-like structure. The 332-Vera genome encodes two proteases and one of them is the head maturation protease. This enzime is highly conservative for the *Chimallinviridae* family and is responsible for the cleavage of many pro-head proteins [50]. Probably, the 332-Vera capsids also undergo proteolytic maturation.

The genes encoding structural proteins are grouped into several clusters in the 332-Vera genome. The largest cluster contains the genes of baseplate protein, portal protein, five tail fiber proteins and others; the sixth tail fiber protein gene is distant from this gene cluster (Figure 3A). Notably, in the 332-Vera genome this large cluster of structural genes contains eight genes of proteins belonging to the radical SAM superfamily. The presence of a large number of radical SAM protein genes in the phage genome is highly unusual [51].

ViPTree analysis confirmed that phage 332-Vera belongs to the new *Chimallinviridae* family (Figure 5A). This phage is a member of a heterogeneous cluster of phages infecting various Gram-negative bacteria. Notably, 332-Vera, infecting *A. veronii*, is more distant from the cluster of other *Aeromonas* giant phages than from the *Pseudomonas* phage phiKZ [48] (Figure 5A). A pairwise comparison of the 332-Vera genome with two most similar genomes of the *Klebsiella* phage vB_Aves_KvM-Eowyn (NC_070650) and *Pseudomonas* phage pPa_SNUABM_DTO1 (MW735835) showed that similarity of all their genes is low and there are differences in gene synteny, including a large insertion (~80 kb) in the 332-Vera genome (Figure 5B). However, the search for core genes using the CoreGenes program (https://coregenes.ngrok.io/, accessed on 15 November 2024) showed that phages 332-Vera, *Pseudomonas* phage pPa_SNUABM_DT01 and *Klebsiella* phage vB_KvM-Eowyn have 68 common genes (E value < 1 × 10^−5^).

Phylogenetic analysis of a number of 332-Vera proteins indicated that dendrograms based on sequences of the terminase large subunit (Figure 5C), major capsid protein, RNA polymerase subunit, tail fiber protein, and portal protein (Appendix A for all sequences) have a similar topology between themselves and ViPTree. However, all protein sequences have not close analogs. Moreover, the sequence of dUTP diphosphatase is related to bacterial analogs, which suggests that the gene encoding this enzyme was relatively recently acquired by phage 332-Vera from the bacterial genome (Figure 5D).

The presence of a large number of long and short tail fiber proteins, sheath proteins, various baseplate proteins in the 332-Vera genome, as well as similar genomic organization of this phage and well-studied *Pseudomonas* phage phiKZ [48], indicated its myovirus morphology. Phage 332-Vera is probably a member of the recently proposed *Chimallinviridae* family, which was confirmed by the presence of genes encoding chimallin and tubulin in its genome. However, a more precise taxonomic identification of phage 332-Vera is difficult, since its genome differs significantly from those of nearby phages: the level of nucleotide identity of this phage genome with the *Pseudomonas* phage pPa_SNUABM_DT01 and *Klebsiella* phage vB_KvM-Eowyn genomes is 36% and 30%, respectively, while the values of intergenomic similarities determined using the VIRIDIC program is 1.6% and 0.6%. The values obtained do not allow these phages to be attributed to the same genus.

**Figure 5 viruses-17-01027-f005:**
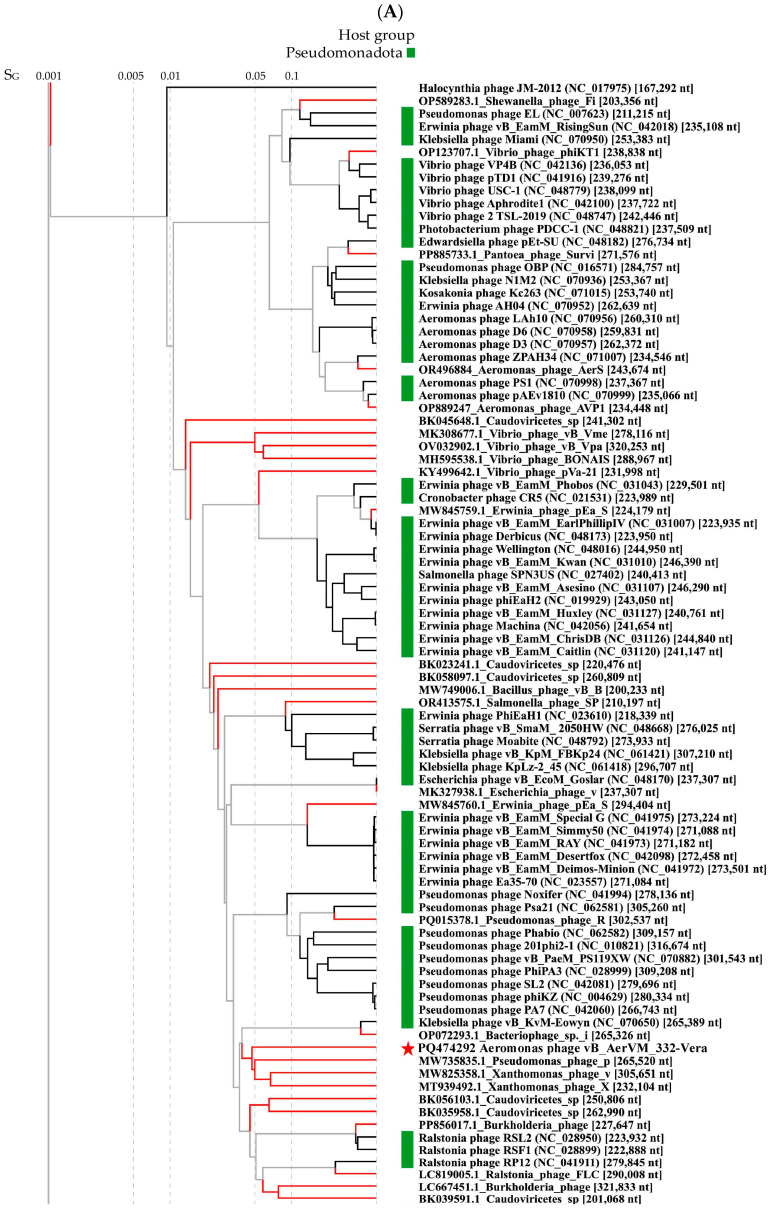
Comparative analysis of the *Aeromonas* phage 332-Vera. (**A**) VipTree analysis of the 332- Vera phage (marked with a red asterisk); *Aeromonas* giant phage AerS_266, previously isolated in Novosibirsk [27] and *Pseudomonas* phage phiKZ [48] are marked with a black asterisk and black triangle, respectively. Phage sequences that were downloaded from the NCBI GenBank manually are marked with red phylogenetic branches. (**B**) A pairwise comparison of the 332-Vera genome and two most similar genomes of the *Klebsiella* phage vB_Aves_KvM-Eowyn (NC_070650) and *Pseudomonas* phage pPa_SNUABM_DTO1 (MW735835), analysis was performed using VipTree software. (**C,D**) Maximum Likelihood phylogenetic tree of the 332-Vera terminase large subunit (**C**) and dUTP diphosphatase (**D**); phylogenetic trees were constructed using the JTT matrix-based LG model in MEGA 11.0 with 1000 bootstrap replicates; studied proteins are marked with black circles.

### 3.7. Comparative Analysis of the 332-Igor Genome

The 332-Igor genome contains genes encoding two DNA polymerase I subunits and three DNA polymerase III subunits; no RNA polymerase gene was found in it, but the gene of phage RNA polymerase sigma factor was found. In addition, the genes encoding six tail fiber proteins, thirteen diverse phage tail proteins, two lysozymes, four peptidases, and lytic transglycosylase domain-containing protein were identified in the 332-Igor genome (Figure 3, Appendix A).

The results of ViPTree analysis indicated that phage 332-Igor is a member of the cluster of giant phages of Gram-negative bacteria. Its genome showed significant differences from those of other phages from this cluster (Figure 6A). A pairwise comparison of the 332-Igor genome with the most similar genomes of *Dickeya* phage vB_DsoM_JA29 (NC_048053) [52] and *Pseudomonas* phage PaBG (NC_022096) [53] confirmed significant differences in the synteny of the 332-Igor genome, for example, the inversion of a genome fragment located at 85,000–110,000 bp and other genomic rearrangements (Figure 6B). There is also a low (if any) similarity of the corresponding genes between the genomes. However, 75 common core genes were found using the program CoreGenes (https://coregenes.ngrok.io/, accessed on 15 November 2024) (E value < 1 × 10^−5^) in the genomes of 332-Igor, *Dickeya* phage vB_DsoM_JA29, and *Pseudomonas* phage PaBG.

Dendrograms based on sequences of the large subunit of terminase (Figure 6C), major capsid protein (Figure 6D), DNA polymerase, tail fiber protein, RecA protein, and portal protein (Appendix A) confirm the distant position of phage 332-Igor.

The obtained results indicate that phage 332-Igor belongs to an unclassified cluster of giant phages. Previously, *Aeromonas* phages have not been discovered in this cluster. The calculated NI levels of the studied phage genome with the genomes of *Pseudomonas* phage PaBG and *Dickeya* phage vB_DsoM_JA29 were 31% and 36%, respectively, while the values of intergenomic similarities determined in the VIRIDIC program were 2.0% and 3.2%. Given such low values, it is possible to conclude that the taxonomic position of phage 332-Igor is uncertain within this cluster of giant phages.

### 3.8. Clusters Containing Giant Phages

Considering the position of the two studied giant phages in different clusters containing giant phages with genomes varying in size and characteristics, all large clusters with both giant and megaphages were analyzed. A total of 848 phage sequences with a genome length of more than 200 kb were extracted from the NCBI GenBank nucleotide sequence database (accessed on 30 September 2024). To remove closely related phages, 148 genomes with a genome similarity index (SG) of less than 0.1 were selected from 848 phage genomes. According to the results of ViPTree analysis, 148 sequences with a genome similarity score (SG) of less than 0.1 were selected, calculated by results of normalized tBLASTx score. Then, 148 selected genomes and two genomes of the studied giant phages 332-Vera and 332-Igor were compared with the ViP Tree database. As a result, ten large clusters (Cluster A–Cluster J) with giant phages were found, having SG with other phages < 0.001 (Figure 7, Appendix A).

The selected ten Clusters differed in the number of phages and the size of their genomes. These Clusters can be further subdivided into 24 sub-clusters (sub-cluster 1–sub-cluster 24) with SG < 0.005 (Table 4). Some of the Clusters and sub-clusters contain only giant phages (A–E, H, and I), whereas some Clusters include both giant phages and phages with smaller genomes (F, G, and J). Thus, sub-cluster 18 of Cluster G is formed by three giant phages (genomes from 203 to 208 kb), belonging to the *Kleczkowskaviridae* family and infecting Gram-negative nodule bacteria. The sub-cluster 18 also contains the phage DSS3_VP1 from the *Naomviridae* family with a short genome, which is close to the ancestral taxon. In addition, crassphages *Cellulophaga* phage phi14:2 and *Flavobacterium* phage vB_FspM from the *Steigviridae* family are close to the sub-cluster 18. Moreover, sub-cluster 19 of the same Cluster G contains a single giant *Acinetobacter* phage vB_AbaM_ME3, whereas all other phages forming this sub-cluster are phages with short genomes. Notably, Cluster H (sub-cluster 20) currently contains only one unique genome sequence Caudoviricetes sp. isolate ct8ad2 from the metagenomic project.

*Aeromonas* giant phages were found in three Clusters D, E, and F. Cluster D (sub-cluster 11) contains a large group of phages belonging to the proposed *Chimallinviridae* family, which includes phage 332-Vera (Figure 5A). Cluster E, sub-cluster 12, did not previously contain *Aeromonas* giant phages and phage 332-Igor is the first *Aeromonas* giant phage in this Cluster (Figure 6A); the genomes of phages forming this Cluster E are mostly 200–268 kb. In addition, a number of *Aeromonas* giant phages were revealed in sub-cluster 17 of Cluster F (Table 4, Appendix A). Other sub-clusters from this large Cluster do not contain *Aeromonas* giant phages.

It should be noted that *Aeromonas* giant phages from different Clusters possibly used a different evolution strategy. Thus, phages from both Cluster D and Cluster E have a similar genome size within the Cluster. However, for phages from cluster D, an ancestral line is known, represented by *Halocynthia* phage JM-2012 with a shorter genome length (~167 kb), which indicates the probable origin of these phages by increasing the size of their genomes. The same was found for Cluster F. Most of phages, forming this Cluster are not giant phages. Probably some of them acquired the genetic material of the hosts and/or other phages, which allowed them to gain some selective advantage and become giant phages including *Aeromonas* ones.

## 4. Discussion

To date, more than 30 species have been described in the *Aeromonas* genus and members of this genus are often found the aquatic environments [54,55]. This makes *Aeromonas* spp. frequent hosts for giant phages, for which aqueous medium is probably favorable due to better diffusion [24]. In this study, three novel *Aeromonas* phages 332-Yuliya, 332-Vera, and 332-Igor and their host *Aeromonas veronii* CEMTC 7594 were found in the same water + sediments sample collected in a freshwater pond, which is located near the Black See, in the Krasnodar Territory. Initially, only phage 332-Yuliya was isolated from this sample after a standard plaque cloning procedure using the host strain *A. veronii* CEMTC 7594. This phage was characterized and it was shown that 332-Yuliya is a member of a group of unclassified siphophages and its genome differs significantly from the genomes of nearby phages.

Considering the possible isolation of giant phages, we used the first confluent plaque to search for these phages. Indeed, two additional genomes were sequenced with good average coverage. One genome, 332-Vera, belonged to the proposed *Chimallinviridae* family, whereas the other genome, 332-Igor, was the first genome of *Aeromonas* giant phages in a large cluster, containing both giant and “small” phages. Notably, both studied giant genomes showed substantial differences from the corresponding nearby genomes.

Attempts to isolate individual giant phages 332-Vera and 332-Igor by plaque assay on the sensitive host were not successful. After inoculation of the *A. veronii* CEMTC 7594 host with a suspension containing a mixture of three *Aeromonas* phages, only 332-Yuliya DNA was identified in all plaques using PCR or real-time PCR; giant phages DNA could not be detected. Our attempts to use 0.3% and 0.5% top agar for obtaining plaques of the studied giant phages were unsuccessful. It has been previously described that giant phages do not usually form visible plaques on double agar [48,56,57].

So, the analysis of the optimal temperature for the studied *Aeromonas* phages growth in liquid culture was carried out by PCR and real-time PCR methods. The obtained results showed that phages 332-Yuliya, 332-Vera, and 332-Igor prefered to grow at different temperatures, whereas their host *A. veronii* CEMTC 7594 propagated at all tested temperature. The water + sediment sample was taken from a pond where the temperature varies significantly depending on the season; in winter, the temperature can drop to 4 °C, and in summer it can rise above 30 °C. When these different phages got into one small pond, they evolved to “exploit” their host in different seasons of the year and thus not to eradicate a single host at all. In turn, the presence of several different bacteriophages infecting a single host leads to a more rapid appearance of bacterial clones adapted to coexist with these phages or even resistant to these phages [58]. Notably, another strategy for the adaptation of *Aeromonas* phages in a closed reservoir has been described previously [59]. In the study, three closely related *Aeromonas* podophages evolved to use different hosts.

We evaluated possible strategies for coevolution of two giant phages 332-Vera and 332-Igor with one host (when there is another phage specific to the same host). An analysis of the evolution of all currently known giant phages was carried out. To isolate various subclusters of giant phages, the cut-off value of the minimum SG, estimated in the ViPTree program, was used as 0.005. The isolation of a new phage family is a complex process that is based not only on the values of genetic similarity, but also on the conservatism of gene synteny, the use of common mechanisms of replication and packaging of the genome into a capsid, and other common features of phages. However, the minimal SG value may be a preliminary basis for analyzing giant phage taxonomy. Thus, ViPTree analysis of the complete tree showed that phages of the *Kyanoviridae* (e.g., *Synechococcus* phage S-MbCM100*), Ackermannviridae* (e.g., *Salmonella* phage BSP101), and *Pootjesviridae* (e.g., *Agrobacterium* phage OLIVR5) families have the SG value of more than 0.01. The use of a more stringent criterion, such as SG < 0.005, can be a preliminary indicator of the allocation of a hypothetical viral family.

Previously, attempts have already been made to taxonomically systematize all giant phages. Ten clusters of giant phages were identified using sequences of terminase and major capsid protein [10]. Eleven major clusters were determined using vConTACT2, which calculates viral clusters based on genome similarity and the general use of orthologous genes [19]. In this study ten large Clusters were identified, which were then divided into 24 sub-clusters [19].

The results of proteomic phylogeny suggested that giant bacteriophages have a polyphyletic origin and they could have evolved in various ways. Undoubtedly, the evolution of phages can include both an increase in the genome and its reduction. The capture of new genes can provide a selective advantage to the phage by acquiring new mechanisms of protection against cellular immune systems. However, a significant increase in the genome can lead to a decrease in the efficiency of replication and transcription of the viral genome. It is also necessary that the organization of the capsid ensures the packaging of an enlarged genome. A significant increase in the viral capsid leads to a decrease in the diffusion of the virus in the environment. So, genome reduction can become evolutionarily beneficial for the phage. It is possible that in different periods of evolutionary history, genome reduction prevailed in some phages (sub-cluster 15). For other phages, the main role was played by the capture of additional genetic information (sub-cluster 16). Notably, using the example of subclusters 13, 14 and 15, one can observe a change in the evolutionary strategy from reduction in genomes to their increase and further reduction. Most giant phages are in clusters 1–5 and 9, which contain only phages with large genomes. Most likely, their mechanisms of interaction with the host, replication, transcription and assembly of viral particles are optimal for large phage genomes. Clusters F, G and J are mainly represented by phages with short genomes; however, some of them acquired the genetic material of the hosts and/or other phages.

## 5. Conclusions

A total of 244 *Aeromonas* phages are deposited in the NCBI GenBank database; of them, 53 belong to giant phages. Here, we describe three novel *Aeromonas* phages: siphophage 332-Yuliya and two giant phages 332-Vera and 332-Igor. These three phages and their host *Aeromonas* veronii CEMTC 7594 were found in the same water + sediments sample collected in a freshwater pond. Despite the host strain’s ability to grow at temperatures from 5 °C to 37 °C, phages 332-Yuliya, 332-Vera, and 332-Igor preferred different temperatures for their reproduction and probably used a single host at different seasons of the year. Taking into consideration that a lot of giant phages are *Aeromonas* phages, it is important to carefully prove the absence of such slow-growing phages in preparations used for control of *Aeromonas* infections. *Aeromonas* giant phages 332-Vera and 332-Igor belong to large clusters, phages that maintain the optimal size of their genomes at the same level. Another group of *Aeromonas* giant phages is included in a large cluster containing mainly phages with “short” genomes, which indicates their gradual increase and appearance of giant phages. It is likely that different *Aeromonas* giant phages, like all giant phages, use various evolution strategies.

## Figures and Tables

**Figure 1 viruses-17-01027-f001:**
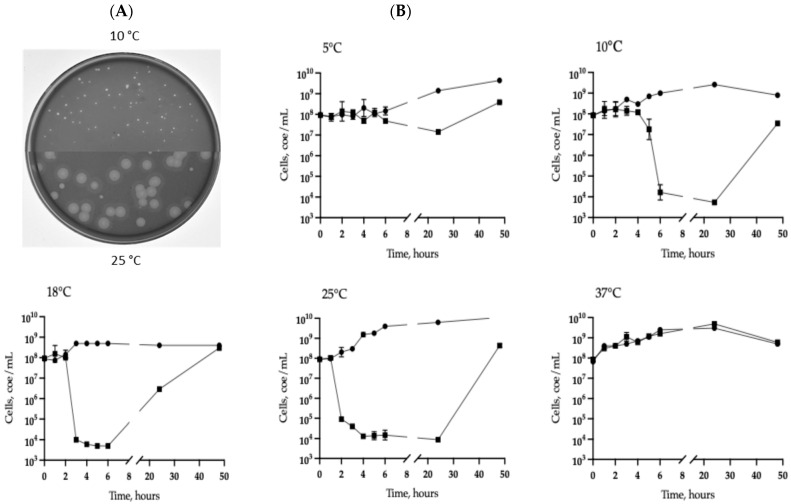
Phage 332-Yuliya characteristics. (**A**) Phage plaques on the lawns of *Aeromonas* CEMTC 7594 obtained after growing at 10 °C or 25 °C. (**B**) Lytic curves for *A. veronii* CEMTC 7594 infected with the phage 332-Yuliya (black boxes) after incubation at different temperatures. Non-infected cultures are marked with black dots. The bars show standard deviation for each point.

**Figure 2 viruses-17-01027-f002:**
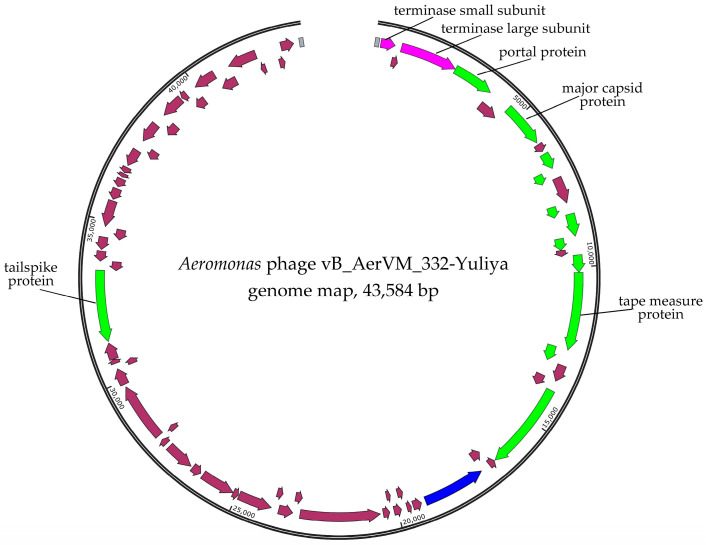
Genome map of the *Aeromonas* phages 332-Yuliya. Genes encoding structural proteins are marked with green arrows; terminase subunits genes are marked with pink; gene encoding DNA polymerase is marked with blue; other genes are brown. Direct terminal repeats (DTRs) are marked with gray boxes.

**Figure 3 viruses-17-01027-f003:**
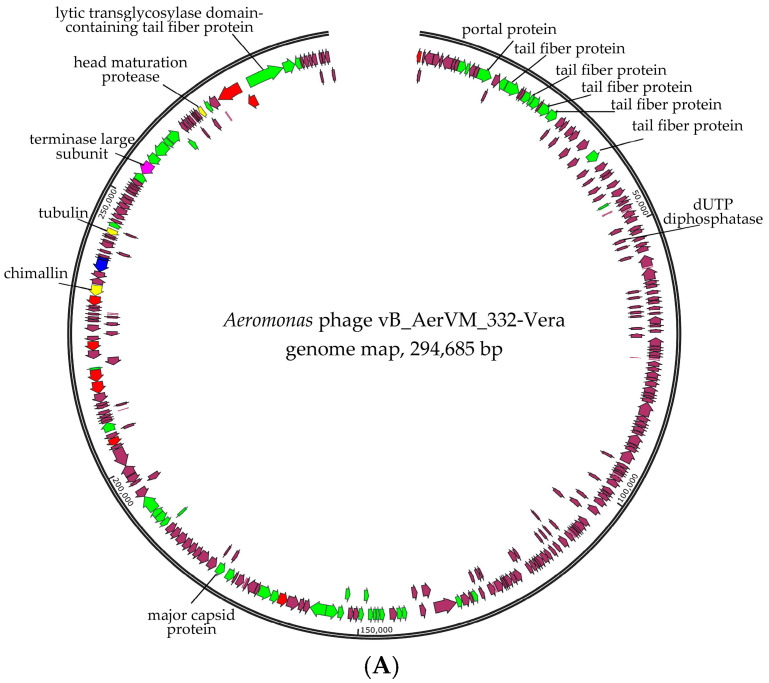
Genome maps of the *Aeromonas* phages 332-Vera (**A**), and 332-Igor (**B**). Genes encoding structural proteins are marked with green arrows; genes encoding RNA polymerase subunits and sigma factor and DNA polymerase subunits are marked with red and blue, respectively; terminase genes are marked with pink; other genes are brown. Genes encoding nuclear shell protein, head maturation protease and tubulin, which are specific for the *Chimallinviridae* family, are marked with yellow in the 332-Vera genome.

**Figure 4 viruses-17-01027-f004:**
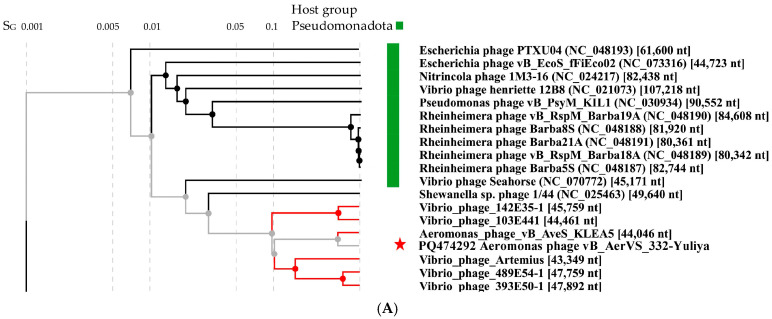
Comparative analysis of the *Aeromonas* phage 332-Yuliya. (**A**) VipTree analysis of the 332-Yuliya phage. The studied phage 332-Yuliya is marked with a red asterisk. Phage sequences that were downloaded from the NCBI GenBank manually are marked with red phylogenetic branches. (**B**) A pairwise comparison of the 332-Yulia genome and two most similar genomes of the *Aeromonas* phage vB_Aves_KLEA5 (OM654374) and *Vibrio*_phage_Artemius (ON366409), analysis was performed using VipTree software. (**C,D**) Maximum Likelihood phylogenetic tree of the 332-Yuliya terminase large subunit (**C**) and major capsid protein (**D)**; phylogenetic trees were constructed using the JTT matrix-based LG model in MEGA 11.0 with 1000 bootstrap replicates; studied proteins are marked with black circles.

**Figure 6 viruses-17-01027-f006:**
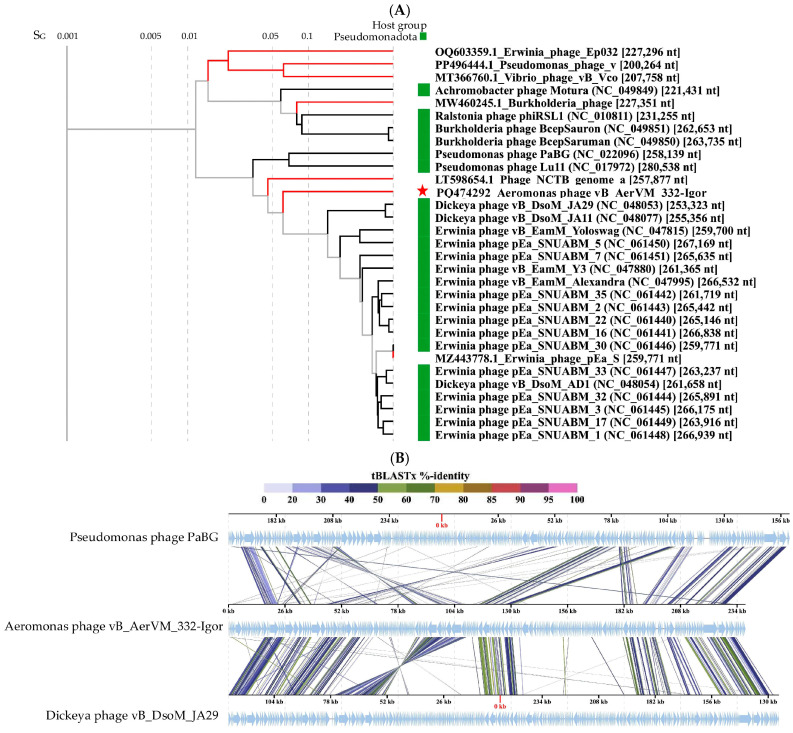
Comparative analysis of the *Aeromonas* phage 332-Igor. (**A**) VipTree analysis of the 332-Igor phage (marked with a red asterisk). Phage sequences that were downloaded from the NCBI GenBank manually are marked with red phylogenetic branches. (**B**) A pairwise comparison of the 332-Igor genome and two most similar genomes of the *Dickeya* phage vB_DsoM_JA29 (NC_048053) and *Pseudomonas* phage PaBG (NC_022096); analysis was performed using VipTree software. (**C,D**) Maximum Likelihood phylogenetic tree of the 332-Igor terminase large subunit (**C**) and major capsid protein (**D**); phylogenetic trees were constructed using the JTT matrix-based LG model in MEGA 11.0 with 1000 bootstrap replicates; studied proteins are marked with black circles.

**Figure 7 viruses-17-01027-f007:**
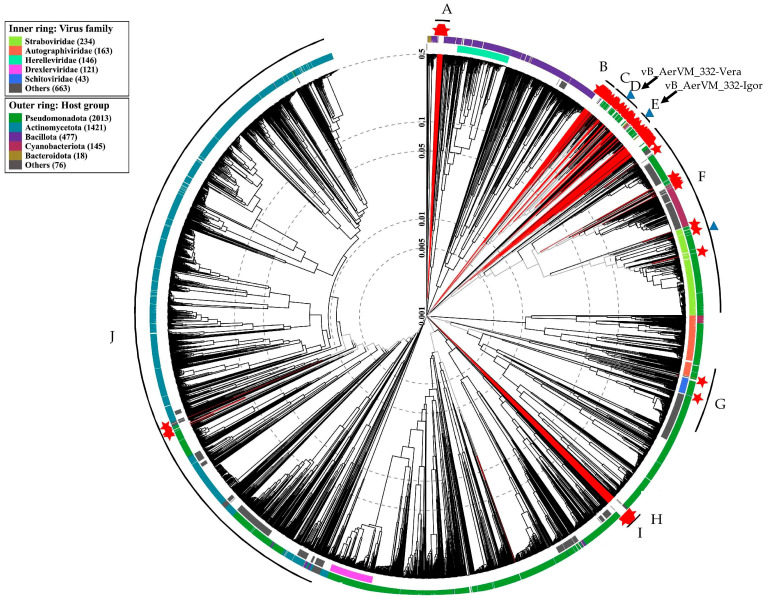
ViPTree analysis (circular tree) for 148 giant phages from the NCBI GenBank database with a genomic similarity score (SG) < 0.1, two studied *Aeromonas* phages 332-Vera and 332-Igor, and 5632 related phages from the ViPTree database. The phages added to the analysis manually are marked with red lines and red stars. The letters show Clusters containing giant and megaphages. Clusters containing *Aeromonas* giant phages are marked with blue triangles. Black arrows indicate Clusters that include phages 332-Vera and 332-Igor.

**Table 1 viruses-17-01027-t001:** Oligonucleotides used for detection and quantification of the ratio of phage genomes.

Phage	Name	Sequence (5’–3’)	Reaction	Product Size (bp)
vB_AerVM_332-Yuliya	43_58U22	CGCTGCTGCTCCTGCTGTCTGG	PCR	646
	43_59_L24	TGCTGAAGCTGGCCTTGAATGGTG		
	43_56U	TCCAGAGCGTATCGACTTCAACGG	Real time PCR	124
	43_56L	ACCGAACTCGTCCACGTCAAAGG		
	43_63P	FAM-ATGCCTACCGAGTCTACAACCGGCAGA-TAMRA		
vB_AerVM_332-Vera	294_58U24	TCTTTGGCTGAGCGTTGGAACACC GGTCCTGACCCATCTTGTCGGACG	PCR	439
	294_59_L24			
	294_55U	ATTGGCTAAGGCTCGCTCTATTGC	Real time PCR	131
	294_56L	CTTCTTTCCAAGTCCAGACCATCCC		
	294_64P	FAM-CCCGCGTTGCCGTGGCTTGTTC-TAMRA		
vB_AerVM_332-Igor	237_59U22	CGACGGGCTGCGGTAAGACGAC	PCR	598
	237_58_L22	ACGGCTTGCAGAATGGACAGCG		
	237_56U	GCTGGTCTGAACACCCATCGTCAC	Real time PCR	134
	237_55L	AGTGGTTGACCGTCTTTGGTCTCG		
	237_63P	FAM-ACGCTGGGAACTCACGTCCGCAGA-TAMRA		

**Table 2 viruses-17-01027-t002:** Characteristics of the 332-Yuliya, 332-Vera, and 332-Igor genomes.

	332-Yuliya	332-Vera	332-Igor
Sequenced genome size (bp)	43,457	294,685	237,907
Physical genome size (bp)	43,584	294,685	237,907
Terminal repeats size (bp)	127	-	-
GC content (%)	47.86	51.46	54.18
Number of predicted genes	68	317	272
Number of predicted tRNA genes	0	0	0
Number of predicted gene products	35	110	99
Number of hypothetical gene products	33	207	173
Positions of the origin of replication (bp)	42,181–42,892	126,966–127,056	210,883–211,002

**Table 3 viruses-17-01027-t003:** Real-time PCR results indicating the presence of phage DNA.

Temperature, °C	vB_AerVM_332-Vera			vB_AerVM_332-Igor			vB_AerVS_332-Yuliya		
	Cq Mean	Cq Std. Dev	E, % *	Cq Mean	Cq Std. Dev	E, % *	Cq Mean	Cq Std. Dev	E, % *
37	-	-	-	-	-	-	30.41	0.025	98.0
25	-	-	-	15.96	0.144	92.9	13.52	0.101	96.5
18	-	-	-	20.40	0.168	94.2	13.48	0.016	98.0
10	30.17	0.334	98.9	28.15	0.093	94.7	13.34	0.266	99.9
5	30.22	0.010	95.0	29.76	0.023	92.4	20.35	0.061	93.0

* E—amplification efficiency; Cq Mean—average quantification cycle value; Cq Std Dev—standard deviation of quantification cycle value.

**Table 4 viruses-17-01027-t004:** Clusters with genomic similarity SG < 0.001, containing giant phages.

Cluster	Subcluster	Contain Only Giant Phages	The Size of Giant Phage Genomes, kb	Contain *Aeromonas* Giant Phages	Known Bacterial Hosts
A	1	+	208–237	-	Gram-positive
	2	+	386	-	-
	3	+	309	-	-
	4	+	227–286	-	-
	5	+	203–227	-	-
	6	+	205–213	-	-
B	7	+	222	-	-
	8	+	229–241	-	Gram-positive and Gram-negative
	9	+	201–288	-	-
C	10	+	236–241	-	Gram-negative
D	11	+	201–321	+	Gram-negative
E	12	+	200–267	+	Gram-negative
F	13	+	447–595	-	-
	14	+	214–235	-	Gram-negative
	15	+	257–735	-	-
	16	+	203–490	-	Gram-negative
	17	-	200–261	+	Gram-negative
G	18	-	203–208	-	Gram-negative
	19	+	235	-	Gram-negative
H	20	+	258	-	-
I	21	+	209–501	-	Gram-negative
	22	+	274	-	Gram-negative
	23	+	208–449	-	Gram-positive and Gram-negative
J	24	-	200–308	-	Gram-positive and Gram-negative

## Data Availability

Genome sequence of the Aeromonas phages vB_AerVM_332-Vera, а vB_AerVM_332-Igor и vB_AerVS_332-Yuliya are available in the GenBank database under the accession numbers: PQ474292, PQ474294, PQ474293, respectively.

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
