# Peer review of "Novel Giant Phages vB_AerVM_332-Vera and vB_AerVM_332-Igor and Siphophage vB_AerVS_332-Yulya Infecting the Same Aeromonas veronii Strain"

_viruses, 2025, doi:10.3390/v17081027_

Round 1
Reviewer 1 Report
Comments and Suggestions for Authors
This manuscript by Babkin et al. describes the single-plaque isolation of one Aeromonas veronii phage and the non-plaque isolation of two others with genomes longer than 200 Kb. The genomic sequences are described and compared to those of other phages. Most of the manuscript does not have new scientific content. Most content is a catalog of more of the same, this time with new phages. Figure 7 and the discussion of it does have new scientific content, but the discussion is without needed details for conclusions about accretive and reductive evolution, for example. The manuscript also has some errors and missing elements. Some obvious missing elements are the details for preparing the two <200 Kb phage DNAs. I am guessing that the top layer gel was blocking the single-plaque isolation of these and even larger phages (see below). The finding of a permuted genome for the two <200 kb phages blocks one of the more interesting enterprises: the deletion of genes non-essential in the laboratory, for phages had been single-plaque isolated (see https://www.tandfonline.com/doi/full/10.4161/bact.19546).
The following are details. An asterisk indicates a key detail.
40: The date of phage G isolation is incorrect. The following is the reference for the isolation. “Donelli, G. Isolamento di un batteriofago di eccezionali dimensioni attivo su B. Megaterium. Cl. Sci. Fis. Mat. Nat. 1969, 44, 95–97.
*41. If long genomes are important here, phiKZ does not have the longest genome in its family. This characteristic goes to phage 201phi2-1, 316,674 bp genome: “Thomas, J.A.; Rolando, M.R.; Carroll, C.A.; Shen, P.S.; Belnap, D.M.; Weintraub, S.T.; Serwer, P.; Hardies, S.C. Characterization of Pseudomonas chlororaphis myovirus 201varphi2-1 via genomic sequencing, mass spectrometry, and electron microscopy. Virology 2008, 376, 330–338.”
*42-44. What is the importance of “jumbo” and “mega”? If none, then these terms should not be introduced.
*47-53. These statements can also be made for phages with genomes smaller than 200,000 bp. If capsid complexity increases with genome size, that is a reasonable point to make. But, adding “jumbo” to the discussion is not productive. And, this point needs an empirical basis. Has a study been made of the genome length-capsid complexity relationship?
*53-57. Good point at the end. However, again, introducing “jumbo” or “giant “is not productive and distracts from the productive point. And, the foundation for this point needs to be more explicitly stated.
*66-71. This paragraph illustrates the main problem. A manuscript needs to be more than a simple database addition.
88-89. Why are the bacteria isolated classified as extremophiles? Sirius appears to be on the Black Sea.
101. Extra “a”.
105. What was the concentration of the top agar? Large phages do not propagate above a critical concentration that becomes lower as the phage becomes larger.
133. More detail is needed.
227-231. How can you eliminate prophages for which you do not have primers? Something is missing in this text.
Figure 1A: Obviously two different Petri plates. However, this should be mentioned.
*Figure 1B. This is too routine to be in the main text.
258. “for” should be “four”.
*268-278. The cloning-isolation sequence for these two >200 kb phages is not presented. It should be presented (in words; more images are not needed). Are the >200 kb phages the ones that made the tiny plaques in Figure 1A? That would fit experience. If so, that means that the upper layer gel had pores too small to isolate even larger phages.
*Figure 3. How are the green (capsid) orfs identified if not one of the standard capsid proteins? I do not see any mass spec data.
*Section 3.5. This section is incomprehensible. What does separation of phages mean? Was the host simultaneously co-infected with all three and PCR used to separately assay the genomes? Otherwise, why wasn’t a plaque assay used? (This point is resolved later in lines 680-689, but should have been clear from the beginning).
*Sections 3.6 and 3.7. More scientific content is needed. Why is any of this of interest?
Figure 7. “The numbers….” should be “The letters….”.
*Figure 7. This is the most science-promoting figure in the manuscript. More discussion is needed of the relevance to phage evolution.
Line 557: Spelling of jumbo.
*Table 4: What phages for a Gram-negative organism have 447-595 Kb genomes? This cannot be presented without being explained.
*565. Speculation that has a very weak basis. Plenty of gels have enough space for > 500 Kb and larger phages (see reference 59).
580-589. Now I understand what was not clear in Section 3.5: The two <200 Kb phages never formed plaques. Better techniques need to be used to do this (see reference 59).
623-624. Other accretive mechanisms potentially exist, especially in the context of microbial community evolution.
620-652 and Figure 7: This is region of the manuscript with, by far, the most science. It should be expanded.
Author Response
We would like to thank the reviewer for his/her comments on our research. We highly appreciate his/her time and effort spent reviewing our paper, and the responses to his/her informative comments have significantly improved the revised manuscript.
This manuscript by Babkin et al. describes the single-plaque isolation of one Aeromonas veronii phage and the non-plaque isolation of two others with genomes longer than 200 Kb. The genomic sequences are described and compared to those of other phages. Most of the manuscript does not have new scientific content. Most content is a catalog of more of the same, this time with new phages. Figure 7 and the discussion of it does have new scientific content, but the discussion is without needed details for conclusions about accretive and reductive evolution, for example. The manuscript also has some errors and missing elements. Some obvious missing elements are the details for preparing the two <200 Kb phage DNAs. I am guessing that the top layer gel was blocking the single-plaque isolation of these and even larger phages (see below). The finding of a permuted genome for the two <200 kb phages blocks one of the more interesting enterprises: the deletion of genes non-essential in the laboratory, for phages had been single-plaque isolated (see https://www.tandfonline.com/doi/full/10.4161/bact.19546).
- We would like to thank the reviewer for carefully reading our manuscript. We believe that our study contains novelty, since all three described phases are at least members of new genera or even subfamilies. In addition, all three phages infect one host, and no other hosts have been found in this pond. These three phages multiply at very different rates, but the optimal temperature for their reproduction varies. Such a coexistence of three phages parasitizing on the same host shows the possibilities of evolutionary adaptation of phages to limited environmental resources, including even slow-growing giant phages.
The following are details. An asterisk indicates a key detail.
40: The date of phage G isolation is incorrect. The following is the reference for the isolation. “Donelli, G. Isolamento di un batteriofago di eccezionali dimensioni attivo su B. Megaterium. Cl. Sci. Fis. Mat. Nat. 1969, 44, 95–97.
- Thank you. This Italian article was not available. This reference is used in the revised version of the manuscript (Ref. 3).
*41. If long genomes are important here, phiKZ does not have the longest genome in its family. This characteristic goes to phage 201phi2-1, 316,674 bp genome: “Thomas, J.A.; Rolando, M.R.; Carroll, C.A.; Shen, P.S.; Belnap, D.M.; Weintraub, S.T.; Serwer, P.; Hardies, S.C. Characterization of Pseudomonas chlororaphis myovirus 201varphi2-1 via genomic sequencing, mass spectrometry, and electron microscopy. Virology 2008, 376, 330–338.”
- Indeed, phiKZ does not have the longest genome in its family. It was noted in the manuscript that this phage was described as a phage with the large capsid (not genome). Just, complete phage genomes can not be sequenced in 1969-1973. Only in the 21st century, the first double-stranded DNA genome with a length of 280 kb of the phage phiKZ was studied.
*42-44. What is the importance of “jumbo” and “mega”? If none, then these terms should not be introduced.
- It has been proposed to name phages with a genome of more than 200 kb “giant or jumbo” phages (Mesyanzhinov et al, 2002; Hendrix et al, 2009; Korn et al, 2021), whereas phages with a genome of more than 500 kb, megaphages (Michniewski et al, 2021). In the revised version of the manuscript, we use the term “giant” as in addition to two giant phages, we describe a phage with the usual genome.
*47-53. These statements can also be made for phages with genomes smaller than 200,000 bp. If capsid complexity increases with genome size, that is a reasonable point to make. But, adding “jumbo” to the discussion is not productive. And, this point needs an empirical basis. Has a study been made of the genome length-capsid complexity relationship?
- Thank you, we correct this inaccuracy in the revised manuscript.
*53-57. Good point at the end. However, again, introducing “jumbo” or “giant “is not productive and distracts from the productive point. And, the foundation for this point needs to be more explicitly stated.
- We add some information for this point to be more explicitly stated
*66-71. This paragraph illustrates the main problem. A manuscript needs to be more than a simple database addition.
- We expand the illustration of the main problem in this paragraph.
88-89. Why are the bacteria isolated classified as extremophiles? Sirius appears to be on the Black Sea.
- We did not classify the bacteria isolate as extremophile. Just the official name of our collection, where all selected microorganisms are deposited, is the Collection of Collection of Extremophile Microorganisms and Type Cultures (CEMTC).
- Extra “a”.
- What was the concentration of the top agar? Large phages do not propagate above a critical concentration that becomes lower as the phage becomes larger.
- At the beginning, 0.8% top agar was used. When the genomes of two additional giant viruses were discovered, attempts were made to obtain plaques formed by the giant phages and isolate them. The host strain A. veronii CEMTC 7594 was inoculated with a phage suspension obtained from the first confluent plaque and five aliquots of the mixture were immediately placed at different temperatures (5 °C, 10 °C, 18 °C, 25 °C, or 37 °C) for phage adsorption (without shaking) and further cultivation (with shaking) at the corresponding temperatures. Then, serial dilutions of the infected cultures, obtained from different temperatures, were spotted on fresh layers of the host strain in the 0.3%, 0.5% and 0.8% top agar and incubated at the corresponding temperature overnight (at 18 °C, 25 °C, and 37 °C) or 48 hours (at 5 °C and 10 °C).
Unfortunately, only plaques, formed by the non-giant siphophage were detected by PCR. Results of Real time PCR confirmed this.
This description is added into the revised manuscript.
- More detail is needed.
- Details are added.
227-231. How can you eliminate prophages for which you do not have primers? Something is missing in this text.
- We did not eliminate prophages. First, we tried to induce them by UV or mytomicin C. No prophages. Second, the obtained genome sequences indicated that these three phages are not lysogenic.
Figure 1A: Obviously two different Petri plates. However, this should be mentioned.
- We clarify this point in the caption of Fig.1.
*Figure 1B. This is too routine to be in the main text.
- We would prefer to keep this figure in the main text as it shows the characteristics of the growth of the host crop at different temperatures, as well as its phage lytic activity at the same temperatures.
- “for” should be “four”.
- Corrected
*268-278. The cloning-isolation sequence for these two >200 kb phages is not presented. It should be presented (in words; more images are not needed). Are the >200 kb phages the ones that made the tiny plaques in Figure 1A? That would fit experience. If so, that means that the upper layer gel had pores too small to isolate even larger phages.
- As plates were incubated at different temperatures (10 °C and 25 °C) overnight (the same time), small clear plaques were observed at 10 °C, whereas two types of plaques were revealed at 25 °C – large plaques with a halo and small clear plaques. To check if different plaques are formed by different phages, both large and small plaques were cloned independently and tested in veronii CEMTC 7594 at 10 °C and 25 °C. However, phage particles independently isolated from small and large plaques formed, in turn, both small clear plaques and large plaques with a halo at 25 °C. The isolated phage was named vB_AerVS_332-Yuliya (332-Yuliya).
- Then, (after genome sequencing) tiny plaques were checked by PCR with specific primers. However, only DNA of the 332-Yuliya phage was found.
*Figure 3. How are the green (capsid) orfs identified if not one of the standard capsid proteins? I do not see any mass spec data.
- Putative open reading frames (ORFs) were determined and annotated using Rapid Annotation Subsystem Technology (RAST) v.2.0 [35] (https://rast.nmpdr.org, accessed on 12 August 2023). The identified ORFs were verified manually using BLAST algorithms against nucleotide and protein sequences, deposited in the NCBI GenBank (https://ncbi.nlm.nih.gov, accessed on 23 September 2023). In addition, the InterProScan [36], HHPred, and HMMER tools [37] were applied for the identification of hypothetical proteins. It is a standard procedure.
*Section 3.5. This section is incomprehensible. What does separation of phages mean? Was the host simultaneously co-infected with all three and PCR used to separately assay the genomes? Otherwise, why wasn’t a plaque assay used? (This point is resolved later in lines 680-689, but should have been clear from the beginning).
- We tried to obtain these phages individually. These experiments are described in the sections Materials and Methods (2.2) and Results (3.5) in the revised manuscript.
*Sections 3.6 and 3.7. More scientific content is needed. Why is any of this of interest?
- It is a common approach to describe main peculiarities of new phage genomes, especially those suggesting new putative taxons.
Figure 7. “The numbers….” should be “The letters….”.
*Figure 7. This is the most science-promoting figure in the manuscript. More discussion is needed of the relevance to phage evolution.
- More detailed description of this Figure is in the Supplementary material.
Line 557: Spelling of jumbo.
- Corrected
*Table 4: What phages for a Gram-negative organism have 447-595 Kb genomes? This cannot be presented without being explained.
В таблице нет phages for a Gram-negative organism with 447-595 Kb genomes
*565. Speculation that has a very weak basis. Plenty of gels have enough space for > 500 Kb and larger phages (see reference 59).
aqueous medium facilitates the diffusion of large phage particles. Это утверждается в большом количестве публикаций, например [24].
580-589. Now I understand what was not clear in Section 3.5: The two <200 Kb phages never formed plaques. Better techniques need to be used to do this (see reference 59).
- Indeed, 0.3% and 0.5% top agar was used in addition to 1% top agar in our attempts to obtain plaques formed by the giant phages. (Just we wanted to look at their electron microscopy!) We add more detailed description in the revised manuscript. These experiments are described in the sections Materials and Methods (2.2) and Results (3.5) in the revised manuscript.
623-624. Other accretive mechanisms potentially exist, especially in the context of microbial community evolution.
- If we talk about the increase in the genome of phages, then the alternative to the capture of new genes from other phages or hosts can only be gene duplication, however, such a mechanism in phages is not common
620-652 and Figure 7: This is region of the manuscript with, by far, the most science. It should be expanded.
- Unfortunately, only genomes of modern phages or MAGs are available. So, only hypothesis or many of them can be suggested. Study of eucaryotic virus evolution is usually based on the genomes isolated in different ages [Babkin I.V., Babkina I.N., Tikunova N.V. An Update of Orthopoxvirus Molecular Evolution Viruses 2022 V. 14 N 2 P. 388 doi : 10.3390/v14020388, Tyumentsev, A. I., Tikunova, N. V., Tikunov, A. Y., Babkin, I. V. Recombination in the evolution of human bocavirus // Infect. Genet. Evol. – 2014. –V. 28 – P. 11–14, Babkin I.V., Tyumentsev A.I., Tikunov A.Y., Kurilshikov A.M., Ryabchikova E.I., Zhirakovskaia E.V., Netesov S.V., Tikunova N.V. Evolutionary time-scale of primate bocaviruses // Infect. Genet. Evol. – 2013. –V. 14 – P. 265–274. Babkin I.V., Tikunov A.Y., Zhirakovskaia E.V., Netesov S.V., Tikunova N.V. High evolutionary rate of human astrovirus // Infect. Genet. Evol. – 2012. – V.12 – P. 435-442.]. This study did not include this work.
Reviewer 2 Report
Comments and Suggestions for Authors
The article submitted for review presents reliable data on the isolation of new bacteriophages with a single host. These data have implications both for expanding the knowledge of bacteriophages and for elucidating pathways for natural control of bacteria in the environment. The results of the bioinformatic analysis of the genomes of the three phages are presented in great detail, which undoubtedly confirm the novelties of this work. Despite the overall good impression, I have a few questions for the authors, especially in the procedures for isolation and characterization of the phages.
1. I think the introduction should be a little more in-depth on the topic
2. The article does not make it clear why prophage induction is performed. Probably the initial ambiguity whether the isolated phages are lytic or lysogenic is the basis of this study. I think this part should be presented more clearly in the Results section
3. Correct the repetition “ this strain this strain” l. 218
4. Which is correct - genomic sequence or genomic sequences – l. 291
5. I cannot find figure S2 in the supplementary files
6. What practical application do you see for your research?
Author Response
We sincerely thank the reviewer for constructive feedback and thoughtful comments, as they have significantly improved the quality of our manuscript. We are confident in the scientific merits of this study.
The article submitted for review presents reliable data on the isolation of new bacteriophages with a single host. These data have implications both for expanding the knowledge of bacteriophages and for elucidating pathways for natural control of bacteria in the environment. The results of the bioinformatic analysis of the genomes of the three phages are presented in great detail, which undoubtedly confirm the novelties of this work. Despite the overall good impression, I have a few questions for the authors, especially in the procedures for isolation and characterization of the phages.
- I think the introduction should be a little more in-depth on the topic
- We add some information in the Introduction section.
- The article does not make it clear why prophage induction is performed. Probably the initial ambiguity whether the isolated phages are lytic or lysogenic is the basis of this study. I think this part should be presented more clearly in the Results section
- We usually perform prophage induction experiments to characterize the host strain, just to be sure that the strain does not contain functional prophage in its genome. We clarified this in the Result section 3.1 (L.224-225). In this particular study, we have to check the presence of the sequenced phage genomes (one belongs to the siphophage) as prophages integrated in the A. veronii CEMTC 7594 genome. We add this information in the Result section 3.5 (L309-312).
- Correct the repetition “ this strain this strain” l. 218
- Corrected
- Which is correct - genomic sequence or genomic sequences – l. 291
- Genome sequences. Corrected.
- I cannot find figure S2 in the supplementary files
- Please, find figure S2 in the supplementary files!
- What practical application do you see for your research?
Taking into consideration that a lot of giant phages are Aeromonas phages, it is important to carefully prove the absence of such slow-growing phages in preparations used for control of Aeromonas infections.